# A Glimpse into Humoral Response and Related Therapeutic Approaches of Takayasu’s Arteritis

**DOI:** 10.3390/ijms25126528

**Published:** 2024-06-13

**Authors:** Shuning Guo, Yixiao Tian, Jing Li, Xiaofeng Zeng

**Affiliations:** 1Department of Rheumatology and Clinical Immunology, Peking Union Medical College Hospital, Chinese Academy of Medical Sciences, Peking Union Medical College, Beijing 100006, China; guoshuningdoct@163.com (S.G.); tyx@alumni.sjtu.edu.cn (Y.T.); 2National Clinical Research Center for Dermatologic and Immunologic Diseases (NCRC-DID), Ministry of Science & Technology, Beijing 100006, China; 3State Key Laboratory of Complex Severe and Rare Diseases, Peking Union Medical College Hospital, Beijing 100006, China; 4Key Laboratory of Rheumatology and Clinical Immunology, Ministry of Education, Beijing 100006, China

**Keywords:** Takayasu’s arteritis, humoral response, autoreactive B cell, rituximab

## Abstract

Takayasu’s arteritis (TAK) manifests as an insidiously progressive and debilitating form of granulomatous inflammation including the aorta and its major branches. The precise etiology of TAK remains elusive, with current understanding suggesting an autoimmune origin primarily driven by T cells. Notably, a growing body of evidence bears testimony to the widespread effects of B cells on disease pathogenesis and progression. Distinct alterations in peripheral B cell subsets have been described in individuals with TAK. Advancements in technology have facilitated the identification of novel autoantibodies in TAK. Moreover, emerging data suggest that dysregulated signaling cascades downstream of B cell receptor families, including interactions with innate pattern recognition receptors such as toll-like receptors, as well as co-stimulatory molecules like CD40, CD80 and CD86, may result in the selection and proliferation of autoreactive B cell clones in TAK. Additionally, ectopic lymphoid neogenesis within the aortic wall of TAK patients exhibits functional characteristics. In recent decades, therapeutic interventions targeting B cells, notably utilizing the anti-CD20 monoclonal antibody rituximab, have demonstrated efficacy in TAK. Despite the importance of the humoral immune response, a systematic understanding of how autoreactive B cells contribute to the pathogenic process is still lacking. This review provides a comprehensive overview of the biological significance of B cell-mediated autoimmunity in TAK pathogenesis, as well as insights into therapeutic strategies targeting the humoral response. Furthermore, it examines the roles of T-helper and T follicular helper cells in humoral immunity and their potential contributions to disease mechanisms. We believe that further identification of the pathogenic role of autoimmune B cells and the underlying regulation system will lead to deeper personalized management of TAK patients. We believe that further elucidation of the pathogenic role of autoimmune B cells and the underlying regulatory mechanisms holds promise for the development of personalized approaches to managing TAK patients.

## 1. Introduction

Takayasu’s arteritis (TAK) is a progressively debilitating form of granulomatous inflammation involving the aorta and its major branches, which is defined by an aberrant immune response to “injury”, leading to intimal hyperplasia and adventitial thickening as well as intramural vascularization [1]. With an incidence of 1–2 cases per million persons estimated in Japan, where TAK was first described [2], recent investigations indicate a broader range, with annual incidence rates spanning from 0.4 to 3.4 per million [3,4,5]. TAK predominantly affects females, with the peak onset typically occurring between the ages of 10 and 40 years. Notably, the pattern of disease differs greatly in terms of sex and age at onset. While the thoracic aorta and its branches are more commonly affected in women, renal and iliac arteritis involvement seem to be particularly prevalent in men [6]. Moreover, younger TAK patients often manifest with active disease accompanied by constitutional symptoms, renal artery engagement and significant ischemic events [7,8,9,10,11]. 

While epidemiological studies have been carried out regarding a broad spectrum of genetic predispositions and environmental cues that may contribute to TAK, the etiology of autoimmune response remains to be elucidated. Emerging data indicate that both innate and adaptive immunity are indispensable for the development and chronicity of vascular inflammation. Although experimental data suggest a dominant role of activated T cells, the incorporation of abatacept (inhibiting T cell activation by blocking the interaction between CD28 and its ligand) into a treatment alongside prednisone failed to decrease relapse risk of TAK patients in a double-blind trial [12]. Notably, a growing body of evidence bears testimony to the widespread effects B cells have on disease pathogenesis and course. B cells are accountable for antibody production, and several serological autoantibodies have been identified in TAK [13]. Furthermore, B cells serve as antigen-presenting cells to the activation of T cells by expressing costimulatory molecules. In particular, immunohistochemical studies of the aortic wall revealed a high proportion of memory B cells and antigen-experienced B cells and the presence of T follicular helper (Tfh) cells in the adventitia of TAK patients [14,15], which produce a broad spectrum of effector cytokines and further participate in tissue destruction and remodeling. Over the past decade, B cell depletion therapy with rituximab (anti-CD20) has shown efficacy in certain cases of TAK [16,17].

Despite the importance of the humoral immune response, a systematic understanding of how autoreactive B cells contribute to the pathogenic process is still lacking. This review aims to elucidate the biological relevance of B cell-derived autoimmunity in TAK pathogenesis and explore therapeutic approaches targeting the humoral response.

## 2. The Role of Autoreactive B Cells

The etiology of TAK is unclear, yet current understanding posits it as an autoimmune disease dominated by T cells [18]. Intriguingly, histological analysis revealed that except T cells, B cells constitute the predominant infiltrating cells within the outer membrane [14]. Initially described by Hoyer and colleagues, the dysfunction of B cells in TAK patients involves an expansion in the population of plasmablasts newly generated in the periphery that related to disease activity [16]. The expansion of plasma cells indicates the overactivity of B cells, a process known to be induced in vitro by interleukin (IL)-6 and B cell activating factor (BAFF) [19,20]. These findings bear striking resemblance to observations in systemic lupus erythematosus (SLE), where B cells play a pivotal role in inflammation through the production of specific autoantibodies. Notably, the inhibition of BAFF levels has demonstrated a marked reduction in expanded plasmablasts, concomitant with clinical improvement [21]. However, the controversial role of B cells in TAK has not been fully explained. Unlike active SLE characterized by lymphocytopenia, no significant decrease in total lymphocyte or B cell counts has been documented in TAK patients in comparison with healthy controls. However, notable alterations of B cell subsets have been observed in active TAK patients. In addition to plasmablast expansion, the data suggest a significantly higher frequency of memory B cells in TAK patients with active disease relative to healthy individuals, coupled with a lower number and frequency of naïve B cells. Conversely, inactive patients exhibited normal naïve B cell counts, with the sole discrepancy being a diminished memory B cell pool [16]. These observations suggest that an earlier introduction of B cell-targeted therapies may hold clinical promise in inducing remission in TAK patients. In addition, the activation factors for autoreactive B cells remain unclear. The loss of immune tolerance precedes a wide array of interconnected abnormal immunological responses. Recent data suggest that products derived from microbes including various bacteria and viruses may induce autoimmunity in TAK through molecular mimicry, where similarities in the sequences of foreign and self-peptides lead to the cross-activation of pathogen-derived autoreactive B cells, culminating in autoimmunity [22]. Given that the immune system of children is not fully developed, they may be more susceptible to excessive immune responses upon microbial challenge, which also contributes to TAK being the third leading cause of vasculitis in the pediatric age group [23]. Work carried out within the past few years has characterized in vitro production of autoantibodies by circulating B cells from patients with TAK [24]. In recent decades, therapeutic interventions targeting B cells, particularly utilizing the anti-CD20 therapeutic antibody rituximab, have demonstrated efficacy in the treatment of TAK [16,17]. These findings suggest a potential involvement of autoreactive B cells in TAK. To date, there remains a paucity of evidence on the subset of autoreactive B cells that evade B cell tolerance checkpoints and the recognition of pathogenic B cell in TAK. In the follicular response, resting naïve B cells undergo differentiation into antibody-secreting cells or into germinal center (GC)-derived switched memory B cells. Notably, recent studies have explored the extrafollicular responses which develop preceding GC formation, presenting distinctive phenotypic and transcriptional profiles compared to GC B cell (Figure 1) [25]. Age-associated B cells (ABCs) are emerging as pivotal constituents of the extrafollicular response, with their aberrant expansion and differentiation increasingly implicated in the pathogenesis of autoimmune ailments. ABCs exhibit a unique phenotype, including classical B cell markers, the transcription factor T-bet and myeloid markers such as CD11c, thus displaying antigen-presenting and proinflammatory capabilities in addition to producing antibodies [26]. Subsequent studies on the extrafollicular activation pathway confirmed that toll-like receptor (TLR) 7 and interferon-gamma (IFN-γ) induce the differentiation of resting naïve B cells into activated naïve B cells, double-negative (DN) 2 cells and antibody-secreting cells in an IL-21 dependent manner in healthy individuals [27]. We notice that several groups have focused on the pathogenic role of the extrafollicular response in TAK, yet the results are unpublished. A deeper comprehension of the distinctive functions of ABCs in immune responses holds significant therapeutic potential with far-reaching implications for effectively targeting these cells, albeit one presenting a formidable challenge.

## 3. Tertiary Lymphoid Organs

“Tertiary lymphoid organs” (TLOs) refers to ectopic lymphoid structures forming during chronic inflammation as a result of lymphoid neogenesis, serving as functional sites for adaptive immune response development. The observations reported so far hint at the existence of TLOs in cancer, infectious or immune diseases, involving synovia in rheumatoid arthritis and salivary glands of patients with Sjogren’s syndrome [28]. TLOs should not be regarded as mere passive bystanders of tissue inflammation due to their effects in promoting autoantibody production and activating cellular effectors, ultimately leading to organ damage. Following selective activation and amplification, B cell clones undergo antibody class switching and somatic hypermutation in GCs within mature TLOs and differentiate into plasma cells to produce antibodies. Moreover, TLOs feature several secondary lymphoid organ structures involving post-capillary high endothelial venules (HEVs), which allow for the homing of naïve cells in T cell areas, creating an interface between T and B cell zones and GC regions [29]. Active TLOs have been identified within the diseased arterial wall of patients with TAK. Initial observations noted lymphoid aggregates in the aortic wall of TAK patients, which posed challenges in differentiation from granulomas [14,30]. However, subsequent studies have effectively characterized the development of peri-aortic lymphoid aggregates in TAK patients and identified active TLOs and granulomas in the diseased vessel wall [15,31]. The similar structures in TLOs and B cell follicles within secondary lymphoid organs indicate a local recruitment mechanism for naïve cells from the periphery via HEV, contributing to establishing a humoral immunological memory. 

Notably, TLOs and granulomas display distinct cellular compositions and occupy separate anatomical niches within the vessel wall (Figure 2). TLOs are distributed deeper in the adventitial layer and feature a dense network of HEVs, while granulomas are located in the medial layer. Several lines of evidence suggest the presence of an antigen-specific immune response in TAK, involving a collaborative interaction between B cells and Tfh cells within the adventitia and TLOs. The presence of memory-like B cells, dendritic cell-like B cells and antigen-experienced Tfh cells in this area may drive an immune pathogenic response, accompanied by long-lived plasma cells and memory CD4^+^ T cells for decades. Furthermore, there is active cross-talk between B cells and T cells in the TLOs within the diseased artery wall of TAK patients. The notable expressions of CXCR5, Bcl-6, PD-1, IL-21, BAFF and CXCL13 are highlighted in TLOs of TAK patients, which play critical roles in B cell activation and the facilitation of B cell–T cell interactions [15]. 

Further, the development and maintenance of TLOs in TAK may rely on a broad spectrum of factors. The Tfh cell compartment, as well as crosstalk between Tfh cells and GC B cells, has been implicated as crucial for TLOs in atherosclerosis-prone mice [32]. Concurrently, IL-6 contributes to B and Tfh cell differentiation, and the inhibition of IL-6 by tocilizumab is effective in TAK [33,34]. The potential inhibitory effect of IL-6 immunotherapy on TLO development in diseased arteries of TAK patients requires further investigation. Collectively, an enhanced understanding of the local immunological responses and the distinctive factors triggering leukocyte aggregation within TLOs will be instrumental in developing novel therapeutic strategies for TAK.

## 4. Autoantibody Production

### 4.1. Non-Specific Autoantibodies

Endothelial cells (ECs) reside at the interface between blood vessels and surrounding tissues, interacting with circulating blood cells, particularly inflammatory cells [35]. Anti-endothelial cell antibodies (AECAs) encompass a diverse array of antibodies targeting ECs, detected across various disorders marked by vascular inflammation including rheumatoid arthritis, anti-phospholipid syndrome, systemic sclerosis, systemic vasculitis and SLE [36]. Since the 1990s, AECAs have been documented in a significant proportion of patients with TAK [37,38,39]. Park and his colleagues identified AECAs in 89.4% of TAK patients, with positive immunoglobulin (Ig)M and IgG AECAs in 83.0% and 68.1% of cases, respectively. Furthermore, they observed a strong correlation between IgM AECA titers and TAK disease activity [40]. While AECAs lack specificity as autoantibodies, they have emerged as playing pathological roles involved in TAK, involving cytotoxicity, EC activation, cytokine production and apoptosis. 

One mechanism involves the cytotoxic impact on ECs, where the interaction of AECAs with ECs occurs through antibody-dependent cell-medicated cytotoxicity (ADCC) or complement-dependent cytotoxicity (CDC). Praprotnik et al. observed that in the presence of a complement, cytotoxicity appears in 58% of AECA-positive TAK sera, which can be enhanced by pretreatment with tumor necrosis factor (TNF)-α or IL-1β, while AECA-negative sera do not [41]. Despite the high prevalence of AECAs in active TAK patients, both active and inactive disease groups exhibit CDC. One plausible explanation is that inactive patients with positive AECAs may harbor histological disease activity not detectable through conventional clinical and laboratory biomarkers employed for assessing disease activity. An autopsy study of Indian TAK patients revealed the presence of active inflammatory arterial lesions during clinically asymptomatic phases of the disease [42]. Possible reasons for the absence of observed ADCC in TAK patients can be delineated as follows. Sgonc et al. reported that ADCC induces EC apoptosis in systemic sclerosis cells via the Fas/Fas ligand pathway [43]. Apoptosis-induced AECAs did not recognize the Fas receptor [44], implying a potential lack of ADCC induction by AECAs in TAK sera. Second, AECA have been implicated in EC apoptosis. Immunoprecipitation studies have underscored the strong recognition of heat shock protein 60 (HSP60) by AECAs [45]. Furthermore, anti-aortic endothelial cell antibodies (AAECA) were detected in 86% of patients with TAK and sera from AAECA-positive patients with TAK elicited a dose-dependent apoptotic response in aortic endothelial cells [46,47]. Moreover, annexin V-mediated apoptosis of ECs and approximately 50% of AECA-positive TAK sera exhibit anti-annexin V activity, supporting the potential of AECAs to induce apoptosis [48]. Third, AECAs trigger the activation of ECs. AECAs derived from TAK patients drive ECs to release an array of effector molecules including IL-4, IL-6 and IL-8, and adhesion molecules such as VCAM-1, ICAM-1 and E-selectin [45,46], thus fostering the inflammatory cascade. Furthermore, Karasawa et al. revealed that increased levels of monocyte chemotactic protein (MCP-1) in TAK patients are associated with antibodies to Prx2, the target antigen of AECAs [49]. The upregulation of MCP-1 suggests not only the recruitment and activation of monocytes but also damage to the small blood vessels of the kidney in TAK patients [50]. Monoclonal AECAs from TAK patients also induced growth-associated oncogene alpha production by ECs and nuclear translocation of nuclear factor kappa-B transcription factor [45,49,51], further significantly augmenting the activation of neutrophil, as well as IL-2, IL-6 and TNF-α genes. Fourth, AECAs exhibit procoagulant effects. Monoclonal AECAs from patients with TAK, when incubated with ECs, induce the production of tissue factor (TF) and initiate the coagulation process [46]. In a consistent manner, the activity, antigens, and mRNA of TF were dose-dependent with AECA titer [52]. Moreover, monoclonal AECAs from TAK patients exhibited increased secretion of von Willebrand factor (vWF) and protein S, enhancing TF binding to factor VII and vWF, thus closing the heparan sulfate on the EC surface to induce thrombosis [45].

In addition to AECAs, numerous other non-specific antibodies have also been identified in TAK. Antiphospholipid antibodies (aPL) are characterized as the principal pathogenic antibodies in antiphospholipid syndrome (APS), which stimulate ECs, thereby inducing a hypercoagulable state [53]. The prevalence of positive antiphospholipid antibodies (aPL) in patients with TAK varies between 0% and 53% [54,55,56,57]. In TAK, aPL might be regarded as an epiphenomenon indicative of the degree of vascular endothelial damage. Additionally, they might play a role in the development of obstructive vasculopathy that emerges subsequent to the initial inflammatory phase. Moreover, several reports have documented the presence of positive anticardiolipin antibodies (ACLAs) [58,59,60], which may play a role in the pathogenesis of vascular injury initiated by endothelial activation and immunological or apoptotic processes [61]. Furthermore, anti-neutrophil cytoplasmic antibodies (ANCAs), which serve to subclassify ANCA-associated vasculitis (AAV), have been reported in TAK. Recent studies indicate that ANCA positivity might be linked to late coronary arterial occlusive disease in TAK patients [62]. Collectively, although non-specific antibodies cannot serve in the diagnosis of TAK, their potential pathogenic role in vascular injury and the association with clinical features yield insights into further understating of TAK.

### 4.2. Specific Autoantibodies

Work carried out has described the presence of antigen-experienced Tfh cells within the inflamed arteries of TAK patients, supporting antigen-driven clonal expansion in TAK [31]. Shirai and colleagues devised an expression cloning system aimed at identifying cell-surface antigens. They utilized a serological autoantigen identification approach, employing both retroviral vectors and flow cytometry [63,64,65]. This endeavor led to the recognition of endothelial protein C receptor (EPCR) and scavenger receptor class B type 1 (SR-BI) as endothelial autoantigens in TAK, which demonstrated robust expression within the vasa vasorum of TAK tissue [66]. The high specificity of anti-EPCR and SR-BI autoantibodies was validated in a cohort comprising 52 active TAK patients and patients with other vasculitis and various autoimmune diseases, with a sensitivity of 67.3% and specificity of 98.0%, which underscore their potential diagnostic utility. EPCR demonstrates cytoprotective and antithrombotic as well as barrier protective properties [67], while SR-BI suppresses endothelial inflammation and cell death [68]. They both serve as negative regulators of inflammation; thus their blocking antibodies exacerbate vascular inflammation within the vasa vasorum. A loss-of-function variant of SR-BI has been linked to the enhanced levels of plasma high-density lipoprotein cholesterol levels and an enhanced risk of coronary heart diseases [69]. Importantly, atherosclerotic lesions are prevalent in TAK, necessitating further elucidation of the precise roles of these antibodies. Meanwhile, anti-EPCR antibodies have been reported to promote Th17 differentiation through binding to activated protein C [66], which further demonstrates the pertinent role of humoral immunity in TAK.

Previous studies highlight the fundamental roles of protein microarray technology, particularly human protein (HuProt) arrays, in identifying and validating autoantibodies as biomarkers for several autoimmune disorders, such as Behcet’s disease and primary biliary cirrhosis [70,71]. In our preliminary research, serum samples were collected from 40 TAK patients, 15 autoimmune disease patients and 20 healthy individuals and subjected to screening by HuProt arrays to identify specific autoantibodies [13]. Subsequently, in an independent cohort, eight candidate autoantibodies were validated, involving anti-QDPR, -PRH2, -SPATA7, -SLC25A2, -ZFAND4, -DIXDC1, -IL17RB and -NOLC1 antibodies. Notably, QDPR exhibited a sensitivity of 71.6% and a specificity of 86.4% in discriminating TAK from healthy and disease controls, while SPATA7 demonstrated a sensitivity of 73.4% and a specificity of 85.4%. SLC25A22 exhibited the highest sensitivity at 80.7%, albeit accompanied by a lower specificity of 67.0%. Moreover, PRH2, NOLC1 and IL17RB exhibited favorable specificities of 88.3%, 86.9% and 85.9%, respectively, albeit with lower sensitivities (<50%). Furthermore, the performance of anti-SPATA7, -QDPR and -PRH2 was validated through Western blot analysis, offering a feasible approach for clinical implementation. Notably, no significant disparity was observed between active and inactive TAK groups, underscoring their utility as diagnostic markers rather than indicators of disease activity. However, the identification of specific autoantibodies has posed challenges owing to the heterogeneity of target antigens. Autoantigens exhibit constitutive expression or undergo translocation from intracellular compartments to cell membranes. While immunoprecipitation or proteomic analyses employing two-dimensional electrophoresis contribute to identifying autoantigens, are limited in their ability to differentiate between cell-surface and intracellular molecules [72]. Additionally, the extraction of numerous membrane proteins presents difficulties in proteomic analysis.

The difficulty for clinical application lies in the lack of specific antibodies for TAK encountered. While laboratory markers of inflammation tests and imaging examinations serve as cornerstones for TAK diagnosis and management, their interpretation can be cumbersome due to their lack of specificity. One plausible reason for this is that patients deemed inactive based on clinical and laboratory assessments may still harbor histological evidence of disease activity. An autopsy study conducted on Indian TAK patients showed the presence of active inflammatory arterial lesions during clinically asymptomatic phases [42]. Even in the same disease, increases in C-reactive protein (CRP) and erythrocyte sedimentation rate (ESR) do not invariably coincide. These routine laboratory markers represent rather crude overall measures. The identification of disease-specific autoantibodies in TAK holds paramount importance for both clinical utility and unraveling the underlying pathophysiological mechanisms. Despite the importance of specific autoantibodies, a dearth of evidence persists regarding their precise role in the pathogenic processes of TAK. Moreover, it is imperative to validate the specificity of these autoantibodies in a larger cohort to facilitate their integration into routine clinical practice.

## 5. B Cell Checkpoints in TAK

It is well established that the regulatory mechanisms governing B cell function, in contrast to T cells, exhibit less stringent control over autoreactivity. The principal sites for the elimination of autoreactive B cells are situated within the bone marrow during early B cell ontogeny, known as central tolerance, followed by subsequent checkpoints within peripheral lymphoid tissues, termed peripheral tolerance [73]. At the central checkpoint, autoreactive B cells undergo receptor editing, subsequent induction of anergy or outright elimination, whereas approximately 20% of mature naïve B cells, which produce antibodies, migrate to the periphery where they may exhibit autoreactivity [74]. Despite documented evidence of dysregulated B cell activation in TAK, a systemic understanding of initial breaches in B cell tolerance remains elusive. In combination with B cell receptors (BCRs) and innate pattern recognition receptors (including TLRs) and co-stimulatory molecules (such as CD40, CD80 and CD86), as well as cytokine receptors collectively influence the developmental fate of individual B cells. Strikingly, genetic variants associated with autoimmunity, identified through genome-wide association studies (GWAS), are highly enriched for signaling programs, involving genes encoding receptors, signaling effectors and downstream transcriptional regulators of BCR, CD40, TLRs or cytokine receptors [75]. Collectively, emerging evidence suggest that dysregulated signaling cascades downstream of B cell receptor families may perturb selection processes, thus biasing naïve B cell repertoires toward autoreactive specificities in TAK (Figure 3). 

BCRs serve as a central regulator orchestrating both negative and positive selection mechanisms, critical for the survival of immature B cells within the bone marrow and mature B cells in the periphery, mediated through nuclear factor-κB-dependent and/or phosphatidylinositol 3-kinase-dependent pro-survival signaling pathways [76]. Notably, our group has identified differentially expressed proteins (DEPs) transported by exosomes in plasma from newly diagnosed TAK and gene ontology analysis of DEPs exhibited the activation of the BCR pathway in TAK [77]. In addition, genetic variants associated with autoimmunity implicated in BCR signaling likely play a role in modulating B cell tolerance. Protein tyrosine phosphatase nonreceptor 22 (PTPN22) exerts negative regulation downstream of BCR signaling. Notably, healthy individuals who harbor the autoimmunity-linked variant of *PTPN22* display heightened autoreactivity within the naïve B cell repertoire, characterized by an increase in transitional and anergic B cell subsets marked by IgD + IgM − CD27 − phenotype [78,79]. In + TAK, the BCR signaling variant *PTPN22^R620W^* is associated with enhanced susceptibility to TAK [80]. While not yet studied in detail, *PTPN22^R620W^* may promote greater autoreactivity in mature cells in TAK. Furthermore, one may propose that the heightened self-reactivity of B cells observed in the pre-immune repertoire of individuals carrying *PTPN22^R620W^* indicates augmented positive selection, rather than diminished negative selection [81]. 

In addition, dual BCR and TLR signaling, facilitated by the transport of self-antigens to endosomal TLRs, orchestrates the initial stimulation of mature naïve autoreactive B cells [82]. TLR7 and TLR9 are responsible for anti-RNA and anti-dsDNA autoantibodies in systemic lupus erythematosus [83]. Our recent study revealed elevated mRNA levels of TLR2 and TLR4 in the TAK group compared to healthy controls [84]. Further research documented that TLR4 activation induced enhanced expression of the IL-1β and IL-1R2 genes in TAK patients [85]. However, the effects of TLR2 and TLR4 on B cells need further confirmation in TAK. CD40 is constitutively expressed on B cells and functions as costimulatory signal. Interactions between CD40 and its ligand (CD40L), with predominant expression on activated CD4^+^ T cells, are indispensable for maintaining proper immune responses, facilitating for GC formation and mediating isotype class switching in response to T-dependent antigens [86,87].

Importantly, dysregulation of CD40 pathway components has been observed in various autoimmune diseases [88,89]. The immunofluorescence of the carotid arterial tissue of TAK patients showed strong expression of CD40 and CD40L [90]. In our previous research, TAK patients exhibited decreased CD40 mRNA levels in peripheral blood mononuclear cells, relative to healthy individuals [91]. The data suggest the daily dose of glucocorticoids is inversely correlated with CD40 mRNA levels [92], suggesting that glucocorticoids may inhibit T cell activation through downregulating CD40. Consistently, gene co-expression networks have shown that compared with the active TAK group, inactive TAK groups have a reduced degree of CD40L and CD40 [91]. While the activation status of CD40 and CD40L in TAK requires further confirmation in newly diagnosed patients without using glucocorticoids and immunosuppressants, the combination of CD40 and CD40L may serve as an indicator of disease status and activity. 

B cell activating factor (BAFF) and a proliferation-inducing ligand (APRIL) refer to the tumor necrosis factor superfamily. Through proteolytic cleavage, they transform into active homo-trimers, exerting significant influence on the maintenance of B cell survival and homeostasis [93]. Although BAFF and APRIL share structural similarities and receptors, they perform distinct functions. BAFF is vital for transitional and mature B cell survival, whereas APRIL influences B-1 cell activity, Ig class switching and the size of the peripheral B cell pool [94]. In addition, elevated levels of BAFF in the periphery prompts autoreactive B cell survival as well as autoantibody production [95]. Research indicates that circulating BAFF levels in TAK patients are higher than those in healthy subjects and giant cell arteritis (GCA) patients [15,20]. Furthermore, longitudinal analysis i of serum BAFF levels in six TAK patients revealed elevation during periods of high disease activity, followed by reduction in response to additional treatments, including TNF-blockers [20]. However, in Indian patients with TAK, BAFF levels were neither elevated nor correlated with disease activity or damage [96]. One possible explanation for this discrepancy is the dual function of BAFF in both a membrane-bound and soluble form. Given that membrane-bound BAFF exhibits a 50-fold greater stimulus for B cell survival compared to its soluble counterpart [97], serum soluble BAFF may not accurately reflect processes occurring at the tissue level. Consistently, histologic analysis of aortic samples from TAK patients demonstrated the expression of BAFF [15].

APRIL exists solely in soluble form, and its serum level serves as a more indictive marker of B cell homeostasis. In patients with active TAK, there is a consistent elevation in serum APRIL levels, which stands in contrast to the levels observed in healthy individuals and those with the inactive disease [96]. However, the mechanism and cell types responsible for the increased levels of BAFF and APRIL in TAK are not yet fully elucidated. It is known that myeloid cells can synthesize BAFF and APRIL in response to cytokine stimulation [93]. Inflammatory cells infiltrating the aorta may secrete cytokines and stimulate the elevated production of BAFF in TAK. Recent studies have observed that the activation of neutrophils, which contain BAFF, in TAK patients [98,99] may serve as a source of BAFF when exposure to inflammatory stimuli [100]. Local production of BAFF by neutrophils in TAK may create a positive feedback loop with inflammation, leading to an increase in circulating BAFF, though it may not be a key link.

Collectively, recent cumulative evidence indicates that signaling through TLRs, CD40, BAFF and APRIL with BCR activation defines the mature B cell repertoire. Moreover, cytokines related to humoral immunity including IL2, IL4, IL6, IL9, IL21, IL23 and IFN-γ exhibited enhanced levels in TAK patients compared with healthy individuals [101]. However, the potential contribution of the interaction between these signaling pathways to the risk of autoimmunity in TAK has not yet been thoroughly investigated. The persistent amalgamation of these signals, contingent upon the specific developmental phase and cellular milieu, functions to regulate both the elimination and survival of cells. One piece of evidence suggesting the crosstalk is that the BCR signaling variant *PTPN22^R620W^*, which is associated with enhanced susceptibility to TAK, is associated with enhanced CD40 expression in mature naïve cells and altered B cells [79]. It is suggested that intrinsic B cell pathways may interplay in an autoimmunity-associated genetic approach, leading to enhanced self-reactivity ae well as polyreactive within the pre-immune repertoire.

## 6. Th Cell Profiles in Humoral Immunity and Pathogenicity

Combinations of cytokine milieu, antigen presentation and expression of costimulatory molecules lead to activated CD4^+^ T cells’ differentiation into several functional classes. Recent investigations have unveiled substantial heterogeneity within CD4^+^ T cell populations, delineating subsets characterized by unique cytokine expression profiles and immune regulatory functions, notably including T helper (Th)1, Th2, Th17 and T follicular helper (Tfh) cells [102]. These subsets serve as pivotal regulators of immune homeostasis, contributing not only to cellular immune responses but also to the facilitation of antibody production by B cells. In addition, CD4^+^ T subsets play pertinent roles in tissue destruction in numerous inflammatory and autoimmune diseases, including multiple sclerosis, rheumatoid arthritis, psoriasis and inflammatory bowel diseases. Multiple lines of evidence implicate the activation of Th subsets in driving the inflammatory cascade in TAK. Serum from active TAK patients cocultured with sorted CD4^+^ T cells from healthy individuals triggered a robust expansion of Th1 and Th17 cells [103]. Furthermore, the presence of IL-17A-, IFN-γ- and IL-6-producing T cells within vascular inflammatory infiltrates in TAK patients underscores the significance of Th1 and Th17 immunity in orchestrating both systemic and vascular inflammation in TAK [103]. Moreover, the identification of a distinct Tfh cell signature within circulating and aorta-infiltrating CD4^+^ T cell populations in TAK patients further underscores the involvement of Tfh cells in disease pathogenesis [15]. Furthermore, matrix correlation analysis has revealed associations between Th1, Th17 and Tfh cell subsets and disease relapse in TAK, with treatment utilizing biologic agents demonstrating efficacy in reducing these subsets [104]. Herein, we elucidated the pivotal roles played by Th and Tfh cells in humoral immunity and their potential contributions to the pathogenic mechanisms underlying TAK.

### 6.1. Th1 Response

Although Th1 cells function as critical players in cellular immunity, IFN-γ, recognized as a major Th1 cytokine, drives IgG class switching. Subsequent research has confirmed that IFN-γR and the signal transducer and activator of transcription (STAT)1 pathway in B cells are critical for spontaneous GC formation and autoimmunity [105]. Moreover, Th1 cells bind to CD40 expressed on B cells via CD40L, thereby enhancing B cell activation. A pronounced expansion of Th1 cells was observed in TAK, which correlated with TAK disease activity [103,106]. Data suggest that Th1 signature cytokines are relevant to the pathophysiology of TAK. IL-12, synthesized by B cells, macrophages and dendritic cells, acts as an inducer alongside IL-18 and type I IFN in the Th1 response. While the serum levels of IL-12 are not described as a suitable disease biomarker for TAK [107,108], increased expression of IL-12 has been observed in aortic tissue from TAK patients with active inflammation [108], indicating the involvement of the Th1 response in the pathophysiology of TAK. Moreover, following stimulation of PBMCs with lipopolysaccharide or phorbol myristate acetate, higher mRNA expression of IL-12 was presented compared with HC [103,109]. IFN-γ is considered the major Th1 cytokine, with observations indicating stronger expression of IFN-γ in infiltrating cells within the arterial wall of active TAK patients compared to those in TAK patients in remission and individuals with atherosclerotic diseases [103,108]. A recent study further illustrated significantly higher levels of IFN-γ associated with active disease in TAK compared to the remission state [110]. Consistently, stimulated PBMCs from TAK patients present higher levels of IFN-γ in supernatants and an increased percentage of CD4^+^ T cells expressing IFN-γ [103,109]. It is of note that glucocorticoids suppressed Th1 response in TAK [111,112], and glucocorticoid therapy leads to lower IFN-γ release by PBMC in the supernatant compared with TAK patients without glucocorticoids [103]. Taken together, these findings underscore the involvement of Th1 cells in the pathogenesis of TAK.

### 6.2. Th2 Response

Th2 cells play an active role in promoting humoral immunity by secreting interleukin-4 (IL-4), IL-5 and IL-13, which stimulate B cell proliferation and IgE production. Mechanistically, the key transcription factor GATA-3 drives Th2 differentiation by binding to regulatory elements within the gene locus responsible for Th2 cytokine expression, ensuring proper expression. Multiple lines of evidence suggest a link between Th2 cells and the pathogenesis of TAK. The observation have revealed increased numbers of Th2 cells and elevated serum levels of Th2-associated cytokines involving IL-4 and IL-13 which are positively related to IL-6 levels in patients with TAK [113]. Additionally, TAK patients constitutively express IL-4 mRNA in PBMCs while no expression is found in healthy controls. Furthermore, upon stimulation, PBMCs from TAK patients exhibit a higher IL-4 mRNA expression compared to healthy individuals [109]. In our own investigation, we observed a notable increase in GATA3 mRNA levels in the PBMCs from patients with active TAK compared to those with inactive TAK. Interestingly, the patients with inactive TAK exhibited inverse correlations between TLRs and GATA3 [84]. These findings underscore the intricate nature of Th2 responses in TAK patients. 

Regulatory T cells (Tregs) function to prevent autoimmunity and control inflammation. Recent data have suggested the significant decrease in the absolute number and proportion of peripheral Treg cells in patients with TAK compared with HCs [114]. Interestingly, several pieces of evidence indicate that Treg cells upregulate the expression of the transcription factors IRF4 and STAT3 and secrete Th2-associated cytokines involving IL-4 and IL-13 to acquire Th2-like phenotypes under certain conditions. The reprogramming process towards Th2-like cells elicits Treg cell dysfunction and induces autoimmune disease [115]. In the context of TAK, a recent study has shed light on the decrease in Th2-like Treg cells in TAK, suggesting their potential role in vasculitis [113]. In systemic sclerosis, Treg cell plasticity can emerge into skin tissue from the peripheral blood, produce Th2 cell-associated cytokines and contribute to fibrosis [115,116]. In TAK, one may suggest that Th2-like Treg cells can infiltrate and damage local tissues, thus the Th2-like-Treg cells decrease in the periphery [113]. Further studies concerning Th2-like Treg cell functions in vessel-localized immune disturbances are warranted in TAK pathogenesis. 

### 6.3. Th17 Response

Among Th subsets, Th17 cells refer to an inflammatory subgroup, mainly driving chronic inflammation within tissues, ultimately leading to organ dysfunction [117]. Importantly, autoantibodies in TAK have been documented to facilitate the differentiation of Th17 cells [66], indicating that autoreactive B cells may serve as an upstream mediator of the pro-inflammatory process of Th17 cells. Th17 cells induced by IL-6, IL-1β and IL-23, exhibit heightened pathogenicity in driving autoinflammatory responses and express enhanced amounts of T-bet, IL-17A, IL-17F and IL-22. Moreover, IL-1 and IL-23 are well established as factors that amplify and stabilize effector Th17 cell responses [102]. Numerous pivotal cytokines in the differentiation and pathogenicity of Th17 cells are increased and associated with disease activity involving IL-17, IL-23, IL-6 and IL-1β in TAK patients [103,112,118]. Furthermore, independent associations with disease parameters have been observed for IL-17A, IL-17E and IL-17F [111]. Given their high sequence homology, IL-17A and IL-17F share similar pro-inflammatory effects [119]. Notably, the presence of the *G allele* at the single nucleotide polymorphism *rs763780* in *IL-17F* gene has been significantly associated with TAK in Asian Indian population [120]. Intriguingly, TAK patients with disease onset after 40 years of age present lower IL-17 levels followed by fewer relapses relative to TAK patients who developed disease manifestations before 40 years of age [121]. Considering the absence of age-related changes in IL-17 in healthy individuals [122], this lends support to the notion that a skewed activation of Th17 cells underlies the systemic and vascular manifestations of TAK. Moreover, studies have demonstrated independent associations of IL-17E and IL-23 with prior ischemic events in TAK patients [111]. Given the pro-atherogenic nature of the Th17 response and its contribution to vascular and systemic inflammation in atherosclerotic disease as well as plaque instability [123], one may suppose that the Th17 response may exacerbate the burden of inflammatory lesions or the development of atherosclerotic lesions in TAK patients. The accumulated evidence reviewed herein seems to suggest the aberrant activation of Th17 cells in TAK patients. In a consistent manner, prominent roles for *IL-2* (an essential suppressor of Th17 differentiation), *IL-6* and *IL-12B loci* (encoding the shared P40 subunit of IL-12 and IL-23) have been identified as novel disease susceptibility in TAK populations from North America and Turkey [124,125]. In particular, clinical trials employing antibodies targeting Th17 pathways involving the IL-23/IL-17 axis have achieved considerable efficacy in treating human diseases [126]. Given that little attention had been given to the IL23/IL-17 axis within tissue inflammation mediated by Th17 cells in TAK, it may be rational to work toward mechanisms and therapeutics targeting the IL-23/IL-17 immune pathway. 

Th cells occupy an important role in the immunopathogenesis of vasculitis, particularly in sustaining the granulomatous inflammation characteristic of arterial lesions. The Th1 and Th17 subsets are currently considered to induce two distinct inflammatory pathways. In GCA patients, Th17 immunity is indispensable for orchestrating acute clinical manifestations, whereas Th1 immunity functions in chronic vascular lesions [127,128]. Emerging evidence has substantiated an independent correlation between elevated levels of cytokines integral to the Th17 response and the extent of arterial involvement, as well as a history of ischemic events in inactive TAK patients [111]. These observations suggest that smoldering arterial inflammation, potentially driven by Th17 cells, persists even during periods of clinical quiescence. Furthermore, investigations have revealed that IL-17E and IL-23 demonstrate independent correlations with a history of ischemic events in TAK patients [111], underscoring the significance of these cytokines in the underlying pathological processes. Considering that the Th17 response is pro-atherogenic and contributes to vascular and systemic inflammation in atherosclerotic disease as well as plaque instability [123], it is plausible to postulate that the Th17 response may exacerbate the burden of inflammatory lesions or the development of atherosclerotic lesions in TAK patients. Moreover, glucocorticoids suppress Th1 cytokines and spare Th17 cytokines in TAK patients [111,112]. In a consistent manner, experimental studies employing knockdown models have demonstrated the deficiency of IFN regulatory 4-binding protein, which serves to advance IL-17 and IL-21 production, triggers rapid development of large vessel vasculitis on account of dysregulated synthesis of IL21 and IL-17A [117]. Relative to the Th1 response, Th17 subsets recruited to and residing within the vessel wall elicit persistent inflammatory effects. Investigations have revealed that CCR6 exhibits invasive tissue capabilities, and notably, the CCR6 receptor is predominantly expressed on Th17 cells. Moreover, CCL20, regarded as a ligand of CCR6, serves to advance selective recruitment of T cells and induce arteritis [129]. Considering all of this evidence, the Th17 response appears to be a principal target of TAK. Given the ineffective role of glucocorticoids in the Th17 response, there is an imperative need to work towards therapeutic strategies aimed at the Th17 response. Although an array of biologics involving TNF inhibitors (TNFi) and tocilizumab are listed among treatment options for refractory TAK patients in recently published guidelines, a significant proportion of TAK patients continue to exhibit suboptimal responses to these biological agents [130]. Janus kinase (JAK) inhibitors have been documented to exhibit efficacy in TAK through the suppression of T helper 1 (Th1) and T helper 17 (Th17) polarization, concomitant with an augmentation in Tregs [131]. Our research group has reported that the IL-17 inhibitor, secukinumab, demonstrated therapeutic efficacy in TAK, significantly reducing ESR, CRP and IL-6 levels, comparable to the effects observed with TNFi [132]. Our findings suggest that secukinumab may represent a viable alternative when TNFi or tocilizumab fail to effectively manage the disease. Additional data from extended follow-up studies are required to substantiate the benefits of secukinumab in reducing glucocorticoid dosage and in inducing favorable arterial structural changes. 

### 6.4. Tfh Response

In peripheral blood, circulating Tfh cells in peripheral blood display phenotypic variances compared to their tissue-specific counterparts, which are characterized by heightened expression levels of PD-1 and ICOS [133]. Ciculating Tfh cells can be delineated based on the expression of CCR6 and CXCR3 [134]. Notably, an increase in CD4^+^CXCR5^+^CCR6^−^CXCR3^−^ T cells, indicative of the Tfh17 cell population, has been observed in the peripheral blood of TAK patients in comparison to both GCA patients and healthy individuals [15,104]. Importantly, the heightened expression of Tfh and Tfh17 signatures detected in individuals diagnosed with TAK remained unattributed to disparities in age, sex, or geographical location [15]. Bcl-6 transcription factor through STAT3 signaling induced by IL-6 and IL21 results in polarization to a Tfh effector class [135]. Data indicated that inappropriate synthesis of IL-21 triggered rapid development of a large vessel vasculitis [117]. Serum levels of IL-21, primarily produced by Tfh and Th17 cells, are related to a more extensive arterial involvement in TAK [111], suggesting that Tfh cells may participate in the disease progression of TAK. In our previous study, TAK patients exhibited an increased mRNA level of *Bcl-6* compared to healthy controls. Further analysis showed the activation of TLRs in TAK and a robust correlation between multiple TLRs and Bcl-6, such as direct interactions and regulatory relationships, indicating that TLRs might function as regulators of Tfh cells [84]. Differentially expressed miRNAs from the plasma exosomes of TAK patients including miR-335-5p, miR-21-5p and miR-34a-5p regulate the expression of Bcl-6 [77]. Taken together, this evidence underscore the activation of Tfh cells in TAK, with their transcription factor implicated in the gene regulatory network contributing to the pathogenesis of the disease.

Moreover, in their pro-inflammatory role, Tfh cells serve a pivotal function in facilitating B cell responses, notably by orchestrating GC formation within lymphoid organs. Facilitated by Tfh cells, B cells undergo hyperproliferation, somatic hypermutation within Ig genes, Ig class switching and differentiation into plasma or memory cells. Research utilizing animal models has highlighted the probable implication of Tfh cells in the development of autoimmune conditions. Stimulated by specific chemicals, mice developed spontaneous systemic autoimmunity, which is related to augmented generation of Tfh cells and spontaneous formation of GC structures [136], and further, the suppression of the Tfh response contributes to disease progression [137]. Investigations into the immune cells present in the adventitia of TAK have disclosed a notable proportion of memory and previously antigen-exposed CD4^+^ T cells, alongside cells displaying characteristic markers associated with Tfh cells including CXCR5, Bcl6 and PD-1 [31]. In vitro studies confirmed Tfh cells from TAK patients facilitate B cell multiplication and maturation through the JAK/STAT pathway [15]. Intriguingly, several groups have identified subsets of Tfh cells exhibiting relationships with Th1, Th2 and Th17 cells [138]. There exist a Tfh phenotype in Th17 cells which that presented high Bcl-6 and IL-21 and could induce the development of IgA-secreting GC B cells in Peyer’s patches [139].Consequently, comprehending the biology of Tfh cells and correlating circulating T cell phenotypes with GC responses and disease states are imperative endeavors.

For decades, Th cell subsets have been classified based on their expression of a limited number of specific cytokines. However, high-dimensional single-cell analysis has unveiled that such a categorization, relying solely on single-cytokine-based nomenclature, inadequately captures the complexity and diversity inherent within Th cell populations [140]. For example, newly identified Th17 populations, termed Th17.1 cells, have been demonstrated to be elevated in TAK patients, which secreted IFN-γ (a signature cytokine of Th1 cells) along with Th17-associated cytokines and were related to active disease [106,141]. Several studies have corroborated the resistance of Th17.1 populations to corticosteroids due to the expression of p-glycoprotein in several autoimmune diseases. Furthermore, in Takayasu arteritis (TAK) patients, these populations have been shown to decrease following treatment with corticosteroids and tacrolimus, indicating the potential of targeting Th17.1 as a therapeutic strategy in TAK [106,142]. Recently, Tualzk and colleagues proposed a categorization of Th cell polarization based on the type of assistance they provide [140]. One of them was type 2 immune response, which was described to primarily foster humoral immunity. Consequently, type 2 Th cells encompass not only Th2 and Tfh cells but also Th1 cells, as all have been demonstrated to be indispensable for humoral (type 2) immunity [143]. In summary, it appears prudent to investigate the operational mechanisms between B and T cells, focusing on target cells and key cytokines, to delineate humoral immunity in TAK.

## 7. Therapeutic Approaches Related to Humoral Response 

### 7.1. B Cell Depletion Therapy 

Recently, emerging data have highlighted that B cell depletion therapy achieves promising therapeutic effects in refractory TAK patients (Table 1). Particularly, rituximab, reserved as a chimeric monoclonal antibody targeting CD20-positive B lymphocytes to induce cell-mediated apoptosis, has shown promising results in reducing inflammatory activity. Hoyer and colleagues initially described the dysfunction of B cells in TAK patients, and rituximab interrupted the generation of plasmablasts, precursors to plasma cells. Subsequent relapses coincided with a resurgence in plasmablast numbers [16], underscoring the potential of plasmablast frequency as a biomarker for rituximab treatment efficacy in TAK patients. In a consistent manner, several lines of evidence described the successful treatment of rituximab for TAK cases with significant CRP and steroid sparing responses [17,144,145]. Furthermore, in contrast to prior cases where rituximab was utilized as a treatment option for refractory patients, a TAK patient employed rituximab in combination with prednisolone as a primary immunosuppressive regimen to attain optimal disease management. Considering the patient’s history of bipolar affective disorder, this approach aimed to circumvent prolonged administration of high-dose steroids. Rapid disease control ensued, validated by subsequent radiological assessments confirming disease stability [146]. Conversely, persistent active disease with radiological progression was observed in four out of seven patients where rituximab was administered as the initial biologic therapy (two patients) or in biologic therapy-naïve individuals. Notably, the patient newly diagnosed with TAK and treated with rituximab during the early disease stages exhibited an unfavorable treatment response [147]. The evidence regarding the suitability of rituximab as a first-line biologic therapy appears to present conflicting perspectives. Notably, Kim and his colleagues described a patient diagnosed with autoimmune encephalitis in the context of TAK who exhibited complete neurological symptom resolution following rituximab treatment. Despite the notable clinical improvement, subsequent brain imaging revealed no discernible interval changes, suggesting that rituximab alone may be insufficient to repair the disruption of the blood brain barrier caused by other immune mechanisms. Subsequently, the patient achieved complete radiological remission after receiving five cycles of infliximab (TNFi) [148]. This finding aligns with previous studies indicating that while rituximab may be effective as second- or third-line therapy, it may not be the optimal choice for initial treatment in TAK.

While most patients treated with rituximab had promising outcomes, there are several limitations substantially hampering the accurate assessment of its efficacy, which may help elucidate the discrepancies observed in these findings. Firstly, the retrospective nature of the study outlined in Table 1 introduces inherent biases, compounded by the concurrent administration of other medications, which complicates the evaluation of rituximab’s effectiveness. Secondly, the majority of studies included only small patient cohorts without a control group for comparison. In fact, fewer than half of cases (3/7), previously unsuccessfully treated with conventional immunosuppressant or biologic therapies, presented a response to rituximab after six months of treatment, along with a decrease in acute phase reactants and the lack of disease advancement [147]. Thirdly, there is a lack of information in most cases on the potential glucocorticoid-sparing effects of rituximab. In addition, response criteria were often unspecified, and imaging response criteria were inconsistently reported.

Although the available data supporting the use of rituximab in TAK seem to be limited to drawing definitive conclusions, rituximab may serve as a viable option as a second- or third-line biologic therapy for TAK cases resistant to conventional immunosuppressive regimens or alternative biologic agents. The effectiveness and safety profile of rituximab, along with its possible application as a primary biologic therapy in TAK patients, warrant validation through controlled clinical trials.

### 7.2. IL-6R Inhibitors

IL-6 serves as a soluble mediator, manifesting diverse impacts on immune responses, hemopoiesis, acute phase reactions and inflammation. Various types of immune cells, fibroblasts and endothelial cells represent crucial loci of IL-6 production. Concurrently, IL-6 executes its biological functions by binding to IL-6R (IL-6 receptor) present on a multitude of cell types. Notably, IL-6 plays a pivotal role in humoral immunity, initially characterized as B cell stimulatory factor 2, facilitating the differentiation of activated B cells into Ab-producing plasma cells [154]. In addition, IL-6 plays a role in promoting Tfh differentiation and stimulating IL-21 production, thereby regulating Ig synthesis, particularly IgG4 production [155]. Consistently, data suggest that IL-6 derived from B cells triggers spontaneous GC formation in systemic autoimmune disorders [156]. A cascade of immune diseases involves the dysregulated and continual synthesis of IL-6 [154]. In TAK, IL-6 levels correlate with acute phase reactants and demonstrate infiltration into the arterial wall among patients with active disease [103,150]. Moreover, IL-6-induced autophagy plays a pivotal role in the vascular fibrosis observed in TAK [157].

The efficacy of tocilizumab, a recombinant anti-IL-6R monoclonal antibody, was initially demonstrated by Nishimoto and colleagues in the treatment of TAK [19]. Subsequently, numerous case reports and observational studies have described favorable outcomes in TAK patients, encompassing clinical amelioration, glucocorticoid-sparing effects and a heightened sustained remission rate in comparison with conventional disease-modifying anti-rheumatic drugs [158,159,160,161,162,163], particularly in patients resistant to TNFi [160,161,163]. A comprehensive meta-analysis including nineteen studies, encompassing a cohort of 466 patients, revealed that 79% of treated individuals achieved a partial clinical response, with approximately 76% experiencing a reduction in glucocorticoid dosage [164]. Additional data derived from uncontrolled studies identified clinical response in 85% and angiographic stabilization in 82% [165]. Comprehensive details regarding completed and ongoing randomized controlled studies blocking IL-6R in patients with TAK are delineated in Table 2 and Table 3.

Following the approval of tocilizumab for TAK, it is noteworthy that sarilumab, an alternative monoclonal antibody targeting the IL-6R, has demonstrated efficacy and safety profiles, receiving approval for RA treatment [171,172]. In addition, three additional monoclonal antibodies directed against IL-6, namely sirukumab, olokizumab and clazakizumab, have undergone clinical trials for various autoimmune diseases [172,173,174]. Further investigation into the therapeutic potential of these alternative monoclonal antibodies in the context of TAK holds promise for elucidating novel treatment modalities.

### 7.3. JAK Inhibitors

Type I and type II superfamily cytokines involving the common γ chain family (IL-2, 4, 7, 9, 13 and 15) have been reported to activate the JAK/STAT pathway [175]. Notably, IL-4, IL-9 and IL-13 are known as typical cytokines in type 2 immune responses, primarily orchestrating humoral immunity [140]. Moreover, JAK/STAT participates in the process of Tfh cells assisting B cell proliferation and activation. In vitro, studies have shown that the inhibition of the JAK/STAT pathway with ruxolitinib (anti-JAK1/2) significantly impedes B cell maturation when co-cultured with Tfh cells derived from TAK. Furthermore, ruxolitinib inhibited IL-6 secretion from either B cells or Tfh cells [15]. In our previous research, we assessed the efficacy and safety profile of tofacitinib in five Chinese patients with refractory TAK. Following a 4-week course of treatment with tofacitinib, clinical manifestations indicative of active disease, such as fever, carotidynia, myalgia, arthralgia and iritis abated in four patients [176]. Another study reported the effectiveness of tofacitinib in treating a patient with TAK complicated by refractory ulcerative colitis following the treatment failure of TNF inhibitor and vedolizumab [177]. TAK patients frequently exhibit severe gastrointestinal complications, particularly inflammatory bowel disease (IBD) [178]. Several lines of evidence suggest that these two diseases share pathological lesions characterized by granulomatous inflammation and a significant proportion of genetic background [179]. Additionally, antibodies targeting colon mucosa and aortic tissue have been identified respectively in ulcerative colitis patients and TAK patients [180], and the cross-reactivity of antibodies may underlie both diseases. Despite the scarcity of evidence on the specific association between IBD and TAK, the successful treatment of tofacitinib may help to elucidate the intricate relationships and shared pathogenic mechanisms between these two diseases. So far, tofacitinib has emerged as a recommended therapeutic option for managing TAK patients with refractory disease or complications [181,182,183]. 

Recently, we initially reported that baricitinib, an oral and reversible JAK1/JAK2 inhibitor, reached an overall treatment response in 60% of patients, with no severe adverse events, in a prospective cohort study [184]. Although the results of the observations were promising, it is necessary to conduct double-blinded, randomized clinical studies to validate the efficacy and safety of JAK inhibitors in treating refractory TAK. Information about relevant finished and ongoing clinical trials targeting JAK in patients with TAK is summarized in Table 2 and Table 3.

In addition, upregulated JAK/STAT signals were identified on the periphery of TAK patients [131]. The increased activity of the JAK/STAT pathway could be attributed to a complex interplay of multiple factors. IL-6 serves as the upstream mediator of STAT3, and the IL-6-STAT3 axis plays a contributory role in the pathogenesis of TAK [108,131]. Furthermore, alongside the identification of novel disease susceptibility loci [185,186], *IL-12B* was closely related to vascular damage in TAK [187]. The increased serum level of IL-12 was also confirmed in TAK [188], and IL-12 drives an inflammatory process through the JAK/STAT pathway [189]. Collectively, these findings suggest a rational approach involving JAK inhibitors in the management of TAK.

Compared with other large vessel vasculitis, TAK is characterized by highly enriched Tfh cell-B cell signature, which fosters Ig production in the peripheral blood [18]. Therefore, TAK has emerged as a disease of heightened complexity, where targeting a singular therapeutic pathway may prove insufficient for achieving remission. In the light of this complexity, therapeutic strategies aimed at concurrently inhibiting multiple cytokines, particularly those involved in humoral responses such as IL-6, along with employing JAK inhibitors, appear promising. Additionally, exploring therapeutic modalities targeting B cell differentiation and plasma cells warrants further investigation as potential novel avenues for managing TAK.

## 8. Conclusions 

The remarkable advancements in humoral immunity research have unveiled novel insights into the multifaceted roles played by B cells, Th cells and the molecular pathways governing their aberrant activity in TAK. Examination across various anatomical compartments (i.e., peripheral blood and vessel wall) revealed the activation of B cells and the existence of humoral immunity in TAK. In addition, a growing body of evidence indicates that B cell depletion therapy yields promising therapeutic outcomes for refractory TAK patients. Building upon these findings, we posit that further elucidation of the pathogenic involvement of autoimmune B cells and the underlying regulatory mechanisms holds the potential to facilitate more tailored approaches to managing TAK patients.

## Figures and Tables

**Figure 1 ijms-25-06528-f001:**
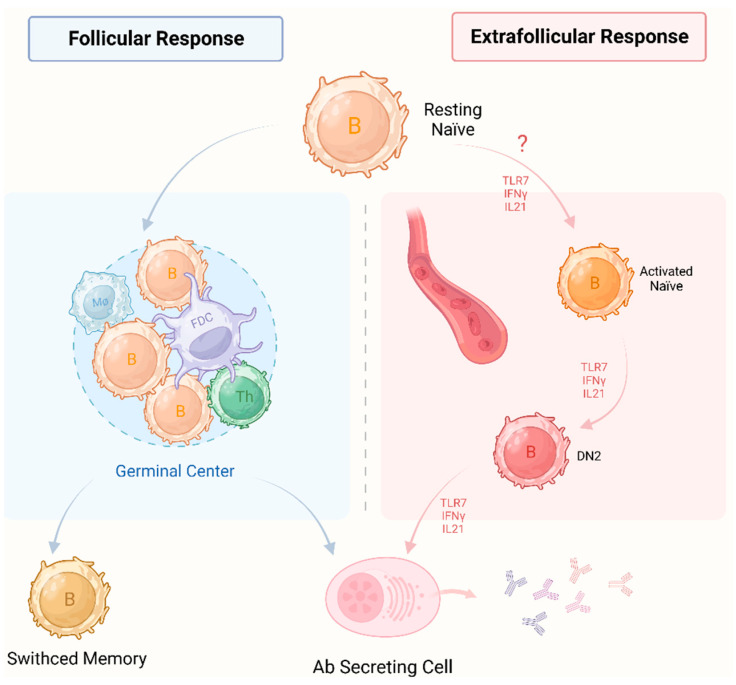
Two pathways of naïve B cells into antibody-secreting cells. In the follicular response (**left**), activated B cells engage in interactions with Th cells and follicle dendritic cells to form GC in secondary lymphoid organs. Following iterative rounds of somatic hypermutation and antigen affinity-driven selection, resting naïve B cells differentiate into antibody secreting cells or switched memory B cells derived from the germinal center. Extrafollicular responses (**right**) emerge preceding the formation of germinal centers, displaying distinctive phenotypic and transcriptional profiles compared to GC B cells. In healthy individuals, TLR7 and IFN-γ induce resting naïve B cells to differentiate into activated counterparts, DN2 cells and antibody-secreting cells in an IL-21-dependent manner. Neither pathway is T cell-dependent. In particular, the extrafollicular response includes a T cell-independent pathway. In addition, both pathways have mainly been reported in systemic lupus erythematosus. In TAK, the pathogenic role of extrafollicular responses is unknown. Therefore, we have marked a question mark on extrafollicular responses. Th: T helper; FDC: follicle dendritic cell; Mø: macrophage; DN2: double negative 2 cells; Ab: antibody; TLR7: toll-like receptor 7; IFNγ: interferon gamma; IL21: interleukin 21; TAK: Takayasu’s arteritis; GC: germinal center.

**Figure 2 ijms-25-06528-f002:**
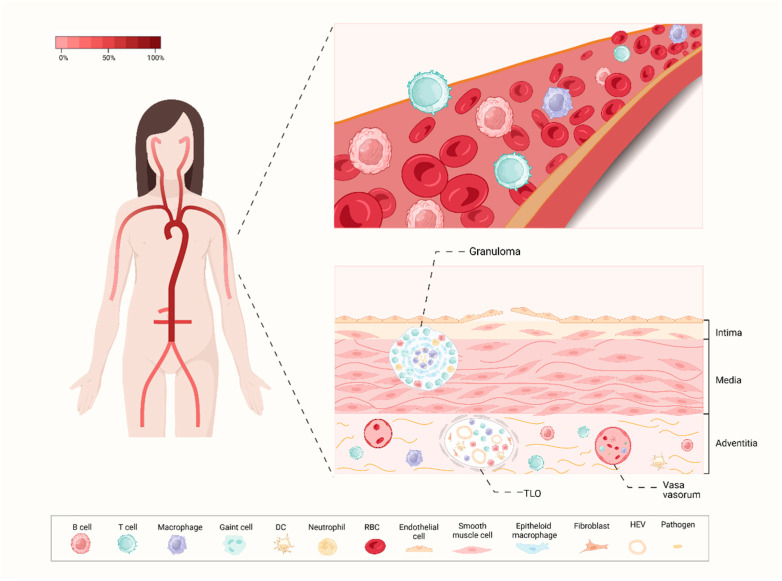
A profile of artery involvement in TAK. In the left part, the color gradient illustrates the typical frequency of arterial segment involvement in TAK, with a predilection for the brachiocephalic arteries, as well as the thoracic and abdominal arterial territories. The right part shows the profile of the peripheral blood and vascular wall of TAK. The pathological process of TAK initiates in the vasa vasorum of the adventitia and is marked by the rupture of elastic laminae and smooth muscle cell migration. Several immune cells including memory B cells, antigen-experienced B cells as well as Tfh cells infiltrate the adventitia. The granulomas are located in the medial layer, and TLOs are distributed deeper within the adventitial layer which involves a dense network of HEVs. TLO: tertiary lymphoid organ; HEV: high endothelial venule; DC: dendritic cell; RBC: red blood cell; Tfh: T follicular helper; TAK: Takayasu’s arteritis.

**Figure 3 ijms-25-06528-f003:**
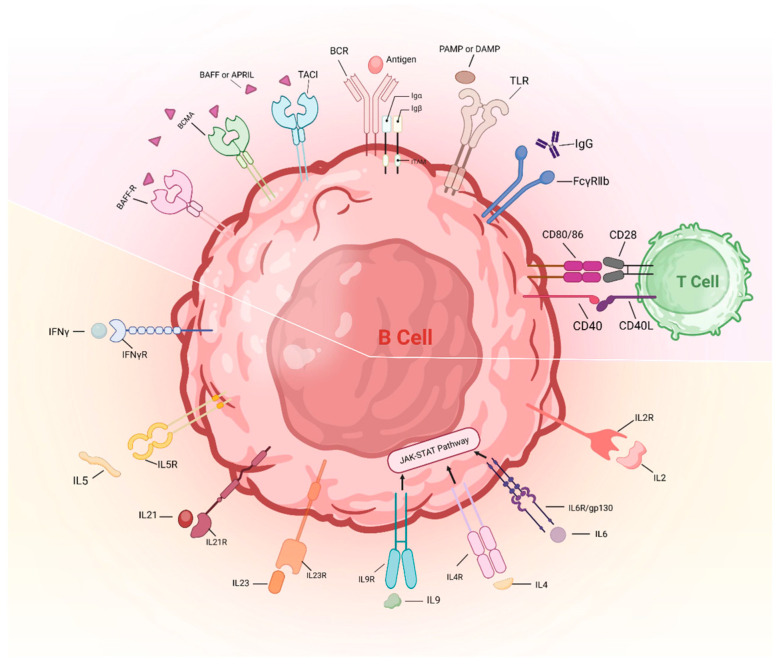
Abnormal activation of B cell checkpoints in TAK. The activation of BCRs, TLRs and several co-stimulatory molecules (including CD40, CD80 and CD86) was documented in TAK. Serum APRIL and BAFF levels and cytokines related to humoral immunity, including IL2, IL4, IL6, IL9, IL21, IL23 and IFN-γ, exhibited enhanced levels in TAK patients compared with healthy individuals. IL-5 induces B cell development and Ig secretion, the role of which is unclear in TAK. The bottom half of the figure is the cytokines and their receptors that are involved in B cell activation. The top half of the figure includes BCRs, TLRs and several co-stimulatory molecules. IL: interleukin; IFNγ: interferon-gamma; R: receptor; BAFF: B cell activating factor; BCMA: B cell maturation antigen; APRIL: A proliferation-inducing ligand; TACI: transmembrane activator and calcium modulator and cyclophilin ligand interactor; BCR: B cell receptor; TLR: toll-like receptor; Ig: immunoglobulin; gp130: glycoprotein 130; TAK: Takayasu’s arteritis; PAMP: pathogen-associated molecular pattern; DAMP: damage-associated molecular patterns.

**Table 1 ijms-25-06528-t001:** Characteristics of the published cases of TAK patients treated with rituximab.

References	Patient Number	Median Age (Years)/ Gender	Median Disease Duration (Months)	Follow-Up Duration (Months)	Therapies with or after RTX	Unsuccessful Therapies before RTX	Clinical Response	Imaging Response
[149]	2	27/2 F	90	NA	None	MTX (2), TNFi (2)	1/2	NA
[16]	3	18/3 F	48	NA	MMF (2), CYC (1)	MTX (2), TNFi (2), MMF (2), CYC (3), HCQ (1)	3/3	1/1 (NA in 2 cases)
[144]	1	25/F	NA	14	CYC	AZA, CYC	1/1	1/1
[145]	1	16/F	3	≥3	MTX	MTX	1/1	NA
[17]	2	32.5/2 F	NA	42 (2)	MMF	CYC (2), MTX (1), ADA (1), IFX (2)	2/2	2/2
[150]	1	42/F	NA	≥6	NA	AZA, CYC, ETN, TCZ	0/1	0/1
[146]	1	39/F	NA	≥36	AZA	None	1/1	1/1
[147]	7	22/1M, 6 F	24	24	MTX (1), MMF (2)	AZA (2), MTX (5), MMF (2), IFX (2), ADA (2), TCZ (2)	3/7	4/7
[151]	8	38/NA	66	12	AZA (2), MTX (1)	AZA (6), CYC (4), MTX (4), MMF (3), CYA (1), IFX: (3), ADA (1)	7/8	NA
[152]	1	34/M	96	36	MTX	MTX, ABT, TCZ, IFX, ETN	1/1	1/1
[153]	2	34.5/F	89.1	NA	MTX (1)	NA	0/2	NA
[148]	1	39/M	11	26	TNFi	CYC	1/1	0/1

ABT: abatacept; ADA: adalimumab; AZA: azathioprine; CYC: cyclophosphamide; ETN: etanercept; HCQ: hydroxychloroquine; IFX: infliximab; MMF: mycophenolate mofetil; MTX: methotrexate; RTX: rituximab; TCZ: tocilizumab; TNFi: tumor necrosis factor inhibitor; NA: not applicable; F: female; M: male; TAK: Takayasu’s arteritis.

**Table 2 ijms-25-06528-t002:** Randomized controlled studies relevant to humoral immune in patients with TAK.

Treatment (Mechanism of Action)	Study Type	Patient Number	Follow-Up Time (Weeks)	Control	Main Results	References
Tocilizumab (IL-6R inhibitors)	Phase 3	36	≥16	Placebo	The results suggested a preference for tocilizumab over placebo concerning the duration until TAK relapse, with no additional safety issues identified, supporting tocilizumab’s effectiveness in managing refractory TAK cases.	[166]
Tocilizumab (IL-6R inhibitors)	Phase 3	28	96	Placebo	It yielded evidence of steroid-sparing effects and enhancements in overall well-being over extended periods of tocilizumab therapy, with no emergence of safety concerns.	[167]
Tocilizumab (IL-6R inhibitors)	Phase 3	28	96	Placebo	Following the initiation of tocilizumab treatment, approximately 60% of TAK patients displayed no advancement in vessel wall thickness. Minimal incidences of dilation/aneurysm progression or stenosis/occlusion were observed among the patient cohort.	[168]
Tofacitinib (JAK inhibitor)	Phase 4	53	36	Methotrexate	Tocilizumab demonstrated superiority over methotrexate in inducing complete remission, displayed a propensity for relapse prevention and facilitate glucocorticoid dose reduction in TAK treatment. The safety of tocilizumab was also confirmed.	[169]
Tofacitinib (JAK inhibitor)	Interventional	67	36	Leflunomide	Tocilizumab and leflunomide were comparable for TAK treatment. Tocilizumab group showed lower side-effect prevalence and higher imaging improvement, as well as a low dose of glucocorticoids to maintain disease remission.	[170]

IL-6R: interleukin 6 receptor; JAK: Janus kinase; TAK: Takayasu’s arteritis.

**Table 3 ijms-25-06528-t003:** Ongoing randomized controlled studies relevant to humoral immune in patients with TAK.

Treatment (Mechanism of Action)	Study Title	Study Type	Current Status	Study Phase	Estimated Completion Date	Clinical Trail Number
Tofacitinib (JAK inhibitor)	Comparison of tofacitinib and prednisolone in the treatment of active Takayasu’s arteritis	Interventional	Recruiting	Phase 3	July 2025	NCT05749666
Upadacitinib (JAK inhibitor)	A study to evaluate the efficacy and safety of upadacitinib in participants with Takayasu’s arteritis	Interventional	Recruiting	Phase 3	August 2027	NCT04161898
Tocilizumab (IL-6R inhibitors)	Multicentre, randomized, prospective trial evaluating the efficacy and safety of infliximab to tocilizumab in refractory or relapsing Takayasu’s arteritis	Interventional	Recruiting	Phase 2	September 2023	NCT04564001
Tocilizumab (IL-6R inhibitors)	A pilot study in severe patients with Takayasu’s arteritis	Interventional	Recruiting	Phase 4	December 2023	NCT04300686

IL-6R: interleukin 6 receptor; JAK: Janus kinase; TAK: Takayasu’s arteritis.

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
