# Peer review of "A Glimpse into Humoral Response and Related Therapeutic Approaches of Takayasu’s Arteritis"

_ijms, 2024, doi:10.3390/ijms25126528_

Round 1

Reviewer 1 Report

Comments and Suggestions for Authors

This is a very comprehensive review on a difficult topic, because Takayasu vascultitis is a rare disease and gathering relevant pathophysiological information is tough. The only missing information in my view, considering the young age of the patients -when the immunological system is just building up from childhood to adolescence and adulthood is whether the authors considered a viral trigger to the whole cascade of immunological disturbances. I have no evidence that this can be the case,  but positive or not, I believe the authors should address this issue in a very short paragraph.

For the rest, congratulations!

Comments on the Quality of English Language

Minor editing issues, mainly typos...

The English language is overall excellent

Author Response

Point-by-point Response to Reviewer’s Comments

We really appreciate you for your carefulness and conscientiousness. Your suggestions are really valuable and helpful for revising and improving our paper. According to your suggestions, we have made the following revisions on this manuscript:

Comments 1: The only missing information in my view, considering the young age of the patients-when the immunological system is just building up from childhood to adolescence and adulthood is whether the authors considered a viral trigger to the whole cascade of immunological disturbances. I have no evidence that this can be the case, but positive or not, I believe the authors should address this issue in a very short paragraph.

Response 1: We appreciate this insightful comment. We notice that microbes including various bacteria and viruses contribute to various autoimmune diseases through the chronic presentation of antigens as a potential source of autoreactivity through molecular mimicry. In Takayasu’s arteritis, recent data suggest that products derived from microbes including various bacteria and viruses may induce autoimmunity through molecular mimicry [1], where similarities in the sequences of foreign and self-peptides lead to the cross-activation of pathogen-derived autoreactive B cells, culminating in autoimmunity. In the revised manuscript, we have complemented that in the section 2 (Page 2-3, line 94-103).

  1. Espinoza, J. L.; Ai, S.; Matsumura, I., New Insights on the Pathogenesis of Takayasu Arteritis: Revisiting the Microbial Theory. Pathogens (Basel, Switzerland) 2018, 7, (3).

Reviewer 2 Report

Comments and Suggestions for Authors

This is an article by S. Guo et al.  

I would like to address a few suggestions to authors that may improve the manuscript.

General recommendations

Please check all the text for spelling mistakes.

Please correct all references according to journal stile.

References should be described as follows:

Journal Articles:

1.      Author 1; Author 2. Title of the article. Abbreviated Journal Name Year; Volume: page range.

Page 2, line 78 and line 93

Please write in vitro in italics.

The controversial role of B cells in TAK has not been fully explained.

Description of tertiary lymphoid organs (TLOs), especially B-cell aggregate-TLOs in the pathogenesis of TAK is better to refer after section 2. Ι suggest to re-write the section 5, the tertiary lymphoid organs before autoantibodies.

The section 3.1. Non-specific autoantibodies, includes predominantly an anti-endothelial cell antibody. Please add same data for other non-specific autoantibodies. Results from previously published studies indicate that antiphospholipid antibodies positivity was not rare in TAK. Moreover, ANCA antibodies are correlated with TAK.

Author Response

Point-by-point Response to Reviewer’s Comments

We really appreciate you for your carefulness and conscientiousness. Your suggestions are really valuable and helpful for revising and improving our paper. According to your suggestions, we have made the following revisions on this manuscript:

Comments 1: General recommendations

Please check all the text for spelling mistakes.

Please correct all references according to journal stile.

References should be described as follows:

Journal Articles:

1.Author 1; Author 2. Title of the article. Abbreviated Journal Name Year; Volume: page range.

Response 1: Thanks for the reviewer’s comment. We have checked all the text for spelling mistakes and corrected all references according to journal style.

Comments 2: Page 2, line 78 and line 93

Please write in vitro in italics.

The controversial role of B cells in TAK has not been fully explained.

Response 2: Thanks for the reviewer pointing this out. We have added this sentence in the revised manuscript (Page 2, line 84-85).

Comments 3: Description of tertiary lymphoid organs (TLOs), especially B-cell aggregate-TLOs in the pathogenesis of TAK is better to refer after section 2. Ι suggest to re-write the section 5, the tertiary lymphoid organs before autoantibodies.

Response 3: Thanks for the reviewer’s wonderful suggestions. We have adjusted the description of tertiary lymphoid organs and corresponding figure after section 2. We also restructured this section and modified some sentences in the revised manuscript (Page 4-6, line 148-202).

Comments 4: The section 3.1. Non-specific autoantibodies, includes predominantly an anti-endothelial cell antibody. Please add same data for other non-specific autoantibodies. Results from previously published studies indicate that antiphospholipid antibodies positivity was not rare in TAK. Moreover, ANCA antibodies are correlated with TAK.

Response 4: We appreciate this insightful comment. We have added recent data for other non-specific autoantibodies in Takayasu’s arteritis and discussed the potential pathogenic role in vascular injury and the association with clinical features in the revised manuscript (Page 7-8, line 258-274).

Reviewer 3 Report

Comments and Suggestions for Authors

1. Figure 1

This figure is almost identical to the figure in Immunol Rev. 2019 Mar;288(1):136-148.  I think it should be noted that both pathwyas are not T cell-dependent. In particular, the extrafollicular pathway includes T cell-independent pathway. Also, in the figure legend, it should be noted that these pathways have mainly been reported in SLE.

2. Autoantibody production 

3.2 specific autoantibodies (Page 5)

I think we should list the representative research results of other researchers before your own preliminary research results.

3. Figure 2

Is the top one a T cell-dependent signaling pathway and the bottom one a T cell-independent signaling pathway? There is no mention of this in the figure legend.

4. Figure 3

I think this chart is very lacking in content. As it stands, it gives us almost no information, so I think it needs to be revised.

5.Treg in TAK

It should be noted that Treg dysfunction in TAK.

Comments on the Quality of English Language

There were several small typos. For example, in Line 23 of the Abstract. Also, "work carried out in the past few years" was repeated over and over again. Please carefully review the entire text and correct it.

Author Response

Point-by-point Response to Reviewer’s Comments

We really appreciate you for your carefulness and conscientiousness. Your suggestions are really valuable and helpful for revising and improving our paper. According to your suggestions, we have made the following revisions on this manuscript:

Comments 1: Figure 1

This figure is almost identical to the figure in Immunol Rev. 2019 Mar;288(1):136-148.  I think it should be noted that both pathwyas are not T cell-dependent. In particular, the extrafollicular pathway includes T cell-independent pathway. Also, in the figure legend, it should be noted that these pathways have mainly been reported in SLE.

Response 1: We are grateful to the reviewer’s critique. The title of the figure has been rewritten and in the revised manuscript, we have explained that both pathways are not T cell-dependent. According to the reviewer’s suggestion, we clarified that the extrafollicular pathway was mainly reported to function in systemic lupus erythematosus, and in Takayasu's arteritis, the pathogenic role of extrafollicular responses is not clear. The details were replenished in the revised manuscript (Page 4, line 130-143).

Comments 2: Autoantibody production

3.2 specific autoantibodies (Page 5)

I think we should list the representative research results of other researchers before your own preliminary research results.

Response 2: We appreciate this insightful comment. We have restructured this section and listed the representative research results at the beginning of this section in the revised manuscript (Page 7-8, line 276-364).

Comments 3: Figure 2

Is the top one a T cell-dependent signaling pathway and the bottom one a T cell-independent signaling pathway? There is no mention of this in the figure legend.

Response 3: Thanks for the reviewer pointing this out. In the revised manuscript, we adjusted Figure 2 to Figure 3 (Page 10, line 386). Because of the large number of cytokines functioned in B cell activation, we included them separately in the bottom half of the figure. For the other scattered factors associated with B cell activation, including BCR, TLR and several co-stimulatory molecules, we have summarized them into the top half of the figure. The T cell-dependent signaling pathway is only one of the scattered factors in the top half of the figure, so we drew this pathway on the top right of the figure. Moreover, according to your suggestion, we have replenished the figure legend and mentioned that in the figure legend (Page 10, line 391-393).

Comments 4: Figure 3

I think this chart is very lacking in content. As it stands, it gives us almost no information, so I think it needs to be revised.

Response 4: Thank you for this comment. We are sorry that the present figure caused a misleading to the readers who might think the figure lacks content. Indeed, the figure we displayed here revealed the complicated pathological and physiological processes of the Takayasu's arteritis, which contained the rupture of elastic laminae, the smooth muscle cell migration, infiltration of multiple immune cells into the adventitia, heterotopia of the granulomas in the medial layer and the formation of TLOs within the adventitial layer during chronic inflammation. In the revised manuscript, we adjusted Figure 3 to Figure 2 (Page 6, line 192). The processes were also described in detail in the figure legend (Page 6, line 193-202). And we earnestly hope that this figure can facilitate better understanding of the vascular involvement and excessively activated immune responses in vascular walls in Takayasu's arteritis.

Comments 5: Treg in TAK

It should be noted that Treg dysfunction in TAK.

Response 5: We appreciate the insightful comments. We have added data describing Treg dysfunction in Takayasu's arteritis in the revised manuscript (Page 15, line 607-609). Given that we aimed to elucidate the humoral response in Takayasu's arteritis, we emphasized the pathogenic role of Th2-like Treg cell (Page 15, line 609-621), which play an active role in promoting humoral immunity by secreting Th2 cytokines.

Comment 6: There were several small typos. For example, in Line 23 of the Abstract. Also, "work carried out in the past few years" was repeated over and over again. Please carefully review the entire text and correct it.

Response 6: Thanks for the reviewer pointing this out. Errors of grammar and expression have been corrected delicately and highlighted in red in the revised manuscript.

Reviewer 4 Report

Comments and Suggestions for Authors

The authors describe the recent discussion of pathogenesis and the new treatment strategy for Takayasu's arteritis (TAK). Overall, this review seems to be well constructed and described. I have only a few minor comments. 

1. JAK inhibitors in TAK: recent case report has shown that a patient with TAK complicated by ulcerative colitis (UC), which already available JAKi for UC treatment, was treated with Tofacitinib (TOF). TOF was effective in both TAK and UC symptoms and this was the first presentation of JAKi for TAK/UC treatment (Sato S, Rheumatology (Oxford) 2020). Please cite this article and add discussion regarding severe gastrointestinal complications, especially UC.

Comments on the Quality of English Language

The Quality of English seems to be adequate. 

Author Response

Point-by-point Response to Reviewer’s Comments

We really appreciate you for your carefulness and conscientiousness. Your suggestions are really valuable and helpful for revising and improving our paper. According to your suggestions, we have made the following revisions on this manuscript:

Comments 1: JAK inhibitors in TAK: recent case report has shown that a patient with TAK complicated by ulcerative colitis (UC), which already available JAKi for UC treatment, was treated with Tofacitinib (TOF). TOF was effective in both TAK and UC symptoms and this was the first presentation of JAKi for TAK/UC treatment (Sato S, Rheumatology (Oxford) 2020). Please cite this article and add discussion regarding severe gastrointestinal complications, especially UC.

Response 1: We appreciate the reviewer's recommendation of this literature which is closely related to the article content. Several lines of evidence suggest that these two diseases share pathological lesions characterized by granulomatous inflammation and a significant proportion of genetic background [1, 2]. The successful treatment of tofacitinib may help to elucidate the intricate relationships and shared pathogenic mechanisms between these two diseases. We have cited this literature and discussed the severe gastrointestinal complications of Takayasu’s arteritis in the revised manuscript (Page 21, line 870-881).

  1. Terao, C.; Matsumura, T.; Yoshifuji, H.; Kirino, Y.; Maejima, Y.; Nakaoka, Y.; Takahashi, M.; Amiya, E.; Tamura, N.; Nakajima, T.; Origuchi, T.; Horita, T.; Matsukura, M.; Kochi, Y.; Ogimoto, A.; Yamamoto, M.; Takahashi, H.; Nakayamada, S.; Saito, K.; Wada, Y.; Narita, I.; Kawaguchi, Y.; Yamanaka, H.; Ohmura, K.; Atsumi, T.; Tanemoto, K.; Miyata, T.; Kuwana, M.; Komuro, I.; Tabara, Y.; Ueda, A.; Isobe, M.; Mimori, T.; Matsuda, F., Takayasu arteritis and ulcerative colitis: high rate of co-occurrence and genetic overlap. Arthritis & rheumatology (Hoboken, N.J.) 2015, 67, (8), 2226-32.
  2. Lee, J. C.; Cevallos, A. M.; Naeem, A.; Lennard-Jones, J. E.; Farthing, M. J., Detection of anti-colon antibodies in inflammatory bowel disease using human cultured colonic cells. Gut 1999, 44, (2), 196-202.

Round 2

Reviewer 3 Report

Comments and Suggestions for Authors

The authors revised the manuscript based on my comments.

The manuscript is now acceptable to this journal.